# The Chromosome-Scale Genomes of *Exserohilum rostratum* and *Bipolaris zeicola* Pathogenic Fungi Causing Rice Spikelet Rot Disease

**DOI:** 10.3390/jof9020177

**Published:** 2023-01-28

**Authors:** Ke He, Chenyu Zhao, Manman Zhang, Jinshao Li, Qian Zhang, Xiaoyi Wu, Shan Wei, Yong Wang, Xujun Chen, Cheng Li

**Affiliations:** 1Key Laboratory of Agricultural Microbiology, College of Agriculture, Guizhou University, Guiyang 550025, China; 2Department of Entomology and MOA Key Lab of Pest Monitoring and Green Management, Plant Pathology Department, College of Plant Protection, China Agricultural University, Beijing 100193, China

**Keywords:** *Bipolaris zeicola*, comparative genomics, *Exserohilum rostratum*, gene prediction and annotation, secretory protein, whole-genome sequencing analysis

## Abstract

Rice spikelet rot disease occurs mainly in the late stages of rice growth. Pathogenicity and biological characteristics of the pathogenic fungus and the infestation site have been the primary focus of research on the disease. To learn more about the disease, we performed whole-genome sequencing of *Exserohilum rostratum* and *Bipolaris zeicola* for predicting potentially pathogenic genes. The fungus *B. zeicola* was only recently identified in rice.We obtained 16 and 15 scaffolds down to the chromosome level for *E. rostratum* LWI and *B. zeicola* LWII, respectively. The length of LWI strain was approximately 34.05 Mb, and the G + C content of the whole genome was 50.56%. The length of the LWII strain was approximately 32.21 Mb, and the G + C content of the whole genome was 50.66%. After the prediction and annotation of *E. rostratum* LWI and *B. zeicola* LWII, we predicted that the LWI strain and LWII strain contain 8 and 13 potential pathogenic genes, respectively, which may be related to rice infection. These results improve our understanding of the genomes of *E. rostratum* and *B. zeicola* and update the genomic databases of these two species. It benefits subsequent studies on the mechanisms of *E. rostratum* and *B. zeicola* interactions with rice and helps to develop efficient control measures against rice spikelet rot disease.

## 1. Introduction

Rice is one of the world’s most important food crops, and the main rice-producing areas are concentrated in Asia; however, rice production and yield are affected by many pathogenic microorganisms found in nature. Two of the most serious rice diseases at present are rice blast caused by the fungus *Magnaporthe grisea* [1] and rice bacterial blight disease caused by the bacterial pathogen *Xanthomonas oryzae* pv. *oryzae* (*Xoo*) [2].However, in recent years, diseases of rice spikes have been increasing in China and are threatening rice production. These include rice brown spot caused by *Bipolaris oryzae*, which produces many brown spots on rice leaves, and rice spikelet rot disease caused by *Fusarium graminearum*, *Alternaria altemata,* and *Nigrospora oryzae*, which initially produces brown spots on rice spikes and white or pink mold on the grain in severe cases [3]. *Exserohilum rostratum* LWI and *Bipolaris zeicola* LWII, the species investigated in this study, also cause rice spikelet rot disease; however, *E. rostratum* LWI has rarely been reported to cause rice disease in China, and *B. zeicola* LWII was recently found on rice glumes for the first time [4].

*Bipolaris* Shoemaker and its related genera are mainly derived from *Helminthosporium* Link. The classification of *Helminthosporium* Link was vague in early studies, and after continuous research, in 1928, according to the shape of the conidia, the relationship of the conidia in the sexual stage, and the mode of germination of the conidia, the *Helminthosporium* Link was classified as *Helminthosporium* Link divided into two subgenera, *Eu-Helminthosporium* and *Cylindro-Helminthosporium*. The genus *Exserohilum* Leonard & Suggs was established by Leonard & Suggs in 1974, and the conidia with protruding umbilical points in *Bipolaris* and *Drechslera* were classified into the genus *Exserohilum* Leonard & Suggs [5]. Although *Exserohilum* is one of the close genera of *Bipolaris*, they can still be distinguished morphologically, the most obvious feature being the distinctive umbilical point of the spores of *Exserohilum* [6]. Particular fungi in the genera *Bipolaris* and *Exserohilum* can cause diseases in animals, plants, and humans. In plants, *E. turcicum* causes northern corn leaf blight (NCLB), a disease that in severe cases kills all foliar tissue, affecting the area of the leaf where photosynthesis takes place, which, in turn, leads to reduced yields [7]; northern corn leaf spot (NCLS) caused by *B. zeicola* is just as harmful as southern corn leaf blight and northern corn leaf blight. A major outbreak of NCLS can result in severe yield and quality losses, affecting leaves, ears, husks, and sheaves of corn [3].In humans, *E. rostratum* causes corneal and skin infections, and the funguscan cross-infect the plant and animal kingdoms [8]. Currently, only *E. rostratum*, *E. mcginnisii,* and *E. longirostratum* are pathogenic to humans [9].

Advances in sequencing technology enable us to further understand the evolution of fungal genomes and provide new insights into the evolutionary history of the eukaryotic community [10]. The genera *Bipolaris* and *Exserohilum* have been progressively studied in recent years, specifically with regard to molecular phylogenetic studies, the pathology of pathogenic infestations, and the determination of genome sequences. The phylogeographic study of the genus and its close genera was established using single- and multigene aggregation analyses. It was found that the constructed phylogenetic tree using ITS-GDPH-spliced sequences would provide a better phylogenetic analysis of the fungi of the genus *Bipolaris.* However, the genera *Bipolaris* and *E. rostratum* and *E. corniculatum* are treated as a polyphyletic group [7,11]. Morphological identification and molecular studies revealed that *E. rostratum* caused leaf blight in ginger, posing a threat to ginger cultivation. Combined with the fact that *E. rostratum* had previously infested plants such as rice and maize, it was hypothesized that many plants would be potential hosts for the fungus [12]. In 2013, the genome sequence of *B. zeicola* 26-R-13 (GCA_000523435.1) was analyzed to obtain 844 scaffolds [13], followed by the determination of 18contigsin *B. zeicola* GZL10 (GCA_016906865.1) from infested maize leaves [14]. *E. rostratum* isolated from a spinal abscess collected from a patient was sequenced and assembled to yield 256 contigs, while the control *E. rostratum* had 1121 contigs [15]. There are few genome assemblies of *E. rostratum* as a plant pathogen. Currently, only *E. rostratum* ZM170581 (GCA_024221855.1) isolated from maize has been reported, with a size of 36.34 Mb. This is the first reported genome sequence of *E. rostratum* isolated from maize [16].

The assembly and annotation of the whole-genome sequences of *B. zeicola* and *E. rostratum* are being carried out and refined. The amount of sequencing data obtained is gradually increasing, speeding up the process of studying the evolution, genetic diversity, and pathogenesis of pathogenic bacteria genomes. However, there are still shortcomings in the current study, such as the use of Illumina data for assembly only, the use of Illumina data and PacBio SMRT data for splicing and assembly is still not optimal, and no whole-genome sequence data of *E. rostratum* on rice. Therefore, in this study, *E. rostratum* and *B. zeicola*, which were isolated from rice glumes for the first time, were used to perform genome sequencing, prediction, annotation, and a comparative genome analysis of the two fungi using PacBio and Illumina high-throughput sequencing technologies to obtain annotated information, including a database of fungal virulence factors and a database of carbohydrate-active enzymes. These results will reveal the functions of rice spikelet rot disease causative genes and provide new directions for elucidating its pathogenesis, as well as important data for genomic studies of *E. rostratum* and *B. zeicola*. In turn, this will lay the foundation for preventing and regulating rice spikelet rot disease.

## 2. Materials and Methods

### 2.1. Collection of Isolates and Genomic DNA Extraction

*Exserohilum rostratum* LWI (Accession number: OQ199492) and *Bipolaris zeicola* LWII (Accession number: OQ199493) were isolated and identified from *Oryza sativa* L. Zhonghua 11 with typical infections collected from the field of the China Agricultural University experimental station in Beijing, kindly provided by Xujun Chen [4]. *E. rostratum* LWI and *B. zeicola* LWII were incubated on potato dextrose medium (PDA) for 7 days at 28 °C. Mycelium was scraped off the surface of the medium with a sterile scalpel, and total genomic DNA of the fungus was extracted using the Biomiga Fungal Genomic DNA Extraction Kit (GD2416, Biomiga, San Diego, CA, USA), and internal transcribed spacer (ITS) sequencing of both flanks was performed. Electrophoresis was performed on a 1% agarose electrophoresis gel, and two-way sequencing was performed by Sangon Biotechnology (Shanghai, China). The ITS sequences of *E. rostratum* LWI and *B. zeicola* LWII were compared with those of the standard strains to determine a match. Afterward, *E. rostratum* LWI and *B. zeicola* LWII were stored in 25% (*v*/*v*) glycerol at 4 °C for subsequent use.

### 2.2. Genome Survey and Repeat Sequence Annotation

JELLYFISH was chosen to assess the genomic heterozygosity, and the genomic heterozygosity was calculated using SOAPaligner/soap2 and SOAPsnp, filtered using the following settings: quality score of consensus genotype ≥ 20, rank–sum test *p*-value >0.05, and minor allele count (supported by ≥5 reads) [17]. Repeat sequences were annotated using RepeatMasker v1.323 [18] and RepeatModeler v1.0.8 (http://www.repeatmasker.org/RepeatModeler/, accessed on 20 June 2022). The genomic sequences were first compared with themselves using RepeatModeler v1.0.8 (parameter setting: -engine ncbi) to construct the repetitive sequence databases for *E. rostratum* LWI and *B. zeicola* LWII, and then RepeatMasker v1.323 (main parameter: -e ncbi) was used for the repetitive sequence analysis. The corresponding results were further counted, resulting in the fasta.sta file as the final statistical result. For de novo gene prediction, we chose Augustus v2.7 [19] and GeneMark + ES v4.0 [20], combining homology and RNA-Seq localization for the protein-coding regions of the *E. rostratum* LWI and *B. zeicola* LWII assemblies. The final gene model was obtained from EvidenceModeler v2012-06-25 [21]. 

### 2.3. Whole-Genome Sequencing and Assembly

*Exserohilum rostratum* LWI and *Bipolaris zeicola* LWII were cultured in potato dextrose broth liquid medium and placed in a shaker at 25 °C and 210 rpm for 5 days. After that, mycelia were extracted and collected in a fume hood using a filter flask. The collected mycelia were frozen with liquid nitrogen and stored in a refrigerator at −80 °C. Genome sequencing was performed on an Illumina HiSeq 2000 system of Novogene (Novogene, Beijing, China), using multiple DNA libraries with pair ends (180 and 500 bp) and mate ends (2, 5, and 10 kb). Trimmomatic v0.32 [22] is used for filtering to obtain high-quality read data. The PacBio Sequel sequencing platform of Novogene was used for whole-genome sequencing, and then, DNA fragmentation was carried out. The BluePippin system was used to recover DNA libraries of more than 20 Kb. After sequencing, the output sequences were filtered using SMRTlinkv5.0 (Pacific Biosciences Technology, Menlo Park, CA, USA) (-minReadScore = 0.8 and -minLength = 1000).For the genome size assessment, we used the software SOAPdenovov2.04 [23] (SOAPdenovo-127mer all -s config.txt -F -K 23 -p 50 -o out_put.), followed by SSPACEv3.0 [24]software to assemble high-quality Illumina reads. The assembled sequences were finally made complete using GapCloserv1.12 [24]. Data from PacBio Seqeul were corrected using Canu v1.5 [25], MECAT v1.3 [26], and NextDenovo v2.3.1(https://github.com/Nextomics/NextDenovo/, accessed on 4 June 2022)/NextPolish v1.3.1 [27]for genome splicing, followed by correction using pilon [28] in combination with Illumina data to improve the accuracy of the PacBio Sequel data. Finally, DBG2OLC [29] was used to mix and assemble the PacBio Sequel and Illumina data. The NextDenovo, Canu, and MECAT splicing results were screened for sequences with only complete 5′ (TTAGGG) and 3′ (CCCTAA) telomeres; after which, the sequences were selected using the MUMmer [30] program for comparative splicing, followed by visualization of the output using mummerplot.

### 2.4. Gene Prediction and Functional Analysis

For de novo gene prediction, we chose Augustus V2.7 [19] and GeneMark + ES V4.0 [20] combined with homology and RNA-Seq sequence localization for the protein-coding regions of the *E. rostratum* LWI and *B. zeicola* LWII assemblies and, finally, EvidenceModeler V2012-06-25 [21] to obtain the gene models. We chose to upload the *E. rostratum* LWI and *B. zeicola* LWII protein sequences to https://international.biocloud.net/, for annotation accessed on 12 July 2022 in the National Center for Biotechnology Information Non-Redundant Protein (Nr) and clusters of orthologous groups for eukaryotic complete genomes (KOG) with the parameters Total or Fungi. We also used the database of protein families (Pfam) for gene function annotation. We chose to do online annotation at http://eggnog-mapper.embl.de/, accessed on 14 July 2022 after TBtools [31] was chosen to analyze the annotation results of Pfam. The OMICSHARE cloud platform (http://www.omicshare.com/tools/Home/Soft/pathwaygsea/, accessed on 14 July 2022) was used to further analyze the protein sequences of *E. rostratum* LWI and *B. zeicola* LWII, the assignment of the reaction to the Kyoto Encyclopedia of Genes and Genomes (KEGG) secondary pathway, and the annotation of the Gene Ontology (GO) database. Protein sequences from *E. rostratum* LWI and *B. zeicola* LWII were uploaded to https://bcb.unl.edu/dbCAN2/blast.php for dbCAN, accessed on 14 July 2022 online annotation with the parameter selection HMMER: dbCAN (E-Value < 1e-15, coverage > 0.35) [32]. Drawing genome circle diagrams and visualizing partial annotation results of *E. rostratum* LWI and *B. zeicola* LWII genomes is possible with circus [33].

### 2.5. Secretome and Effectors Predictionand Toxicity Factor Prediction

Predictions of secreted proteins were made using SignalP v6.0 Server [34] to remove proteins without signal peptides, followed by TMHMM Server v1.0.10 [35] and Phobius v1.01 [36] to take intersections to remove proteins with transmembrane domains, ProtComp v9.0 [37] and WoLF PSORT [38] to take intersections to predict protein positions, and finally, PredGPI [39] to remove anchored proteins. The effector proteins were screened using the Klosterman standard, and the results of the secreted proteins were screened for amino acid numbers less than 300, followed by SnapGene software (https://www.snapgene.com/, accessed on 15 July 2022) to screen for genes with less than four cysteines. Finally, EffectorP 3.0 (https://effectorp.csiro.au/, accessed on 15 July 2022) was used to obtain the final effector proteins. The Pathogen–Host Interaction Database (PHI-base) can be searched at http://www.phi-base.org, accessed on 16 July 2022, and candidate virulence-associated genes were identified using BLASTp against PHI-base v4.3 [40].

### 2.6. Phylogenetic and Homology Analysis

OrthoMCL v2.0.9 [41] was used for the localization and annotation of direct homologs of *E. rostratum* LWI and *B. zeicola* LWII, followed by alignment using all-versus-all BLASTP (E-value ≤ 1 × 10^−5^, coverage ≥ 50%). Orthofinder [42] was used to locate single-copy genes, followed by MAFFT v7.22196 [43], and Gblocks v0.91b [44] was used to extract conserved loci from the alignment results. The phylogenetic tree was constructed using RA×ML v8.1.24 [45] based on the Maximum Likelihood, with *Fusarium graminearum* as the outgroup.

## 3. Results

### 3.1. Pathogen Identification

*Exserohilum rostratum* LWI and *Bipolaris zeicola* LWII cause rice spikelet rot disease, in which the glumes of rice become chlorotic at the initial stage of the disease and brown spots develop over time, reducing rice yield and quality. *E. rostratum* LWI and *B*. *zeicola* LWII isolated from diseased spike parts were incubated on potato agar medium at 28 °C for 7 d in a 12 h photoperiod incubator, after which they were analyzed morphologically. In order to allow *E. rostratum* LWI and *B. zeicola* LWII to produce spores, water agar medium with wheat straw (TWA-W) was used. We used TWA-A medium at 28 °C for 5 d to produce conidia for *E. rostratum* LWI and *B. zeicola* LWII. The colonies of *E. rostratum* LWI were characterized by the production of a large number of white aerial hyphae with a raised center, which gradually turned brown in the center and white on the outer edge. When the conidia were observed under the microscope, they were brown in color and had an elongated oval shape with a central umbilical point at the base, with 3–8 septa and sizes of 48–80 μm × 9–19 μm. In contrast, the colonies of *B. zeicola* LWII had neat, dark grey edges, and the hyphae were white initially, then gradually became dark brown in color in the center. The conidia were observed to be slightly wider in the middle and taper at the ends, with bluntly rounded basal cells, brown in color, with 6–10 septa and sizes of 45–80 μm × 10–15 μm. The strainsLWI and LWII were later identified as *E. rostratum* and *B. zeicola* based on a phylogenetic tree constructed from the ITS and single-copy homologous genes results. *E. rostratum* LWI and *B. zeicola* LWII were inoculated on rice spikes by spray inoculation, respectively, and rice spikes sprayed with both showed typical symptoms, while the control group showed no symptoms (Figure 1A, Figure 2A), satisfying Koch’s hypothesis. Reisolation of the pathogenic fungi from infested rice spikes confirmed that these symptoms, which had the same morphological characteristics as the original pathogen, were consistent with the characteristics described for *E. rostratum* LWI and *B. zeicola* LWII.

### 3.2. Genome Sequencing and Assembly

Based on Illumina reads, we sequenced and assembled *E. rostratum* LWI and *B. zeicola* LWII, and all high-quality data from *E. rostratum* LWI and *B. zeicola* LWII were evaluated using the k-mer analysis. The expected depths of the k-mers correspond to sequencing depths of 122x and 93.8x, respectively, and their sizes were estimated by the k-mer analysis. *E. rostratum* LWI was estimated to be 34.909 Mb with 93.8% non-repeated sequences, while *B. zeicola* LWII was 40.011 Mb with 74.2% non-repeated sequences (Appendix A). Afterward, Novogene’s PacBio Sequel sequencing platform was used for the whole-genome sequencing of *E. rostratum* LWI and *B. zeicola* LWII; the PacBio Sequel and Illumina data were mixed, and DBG2OLC was used for assembly. Finally, both *E. rostratum* LWI and *B. zeicola* LWII were assembled at the chromosome level, with 16 and 15 chromosomes, respectively. There were putative telomeric repeats 5′-(TTAGG)n-3′ at both their F-terminal and R-termini. After high-quality control, the total sizes of *E. rostratum* LWI and *B. zeicola* LWII assembled were 34,053,972 and 32,215,838 bp, the N50 lengths were 2,207,071 bp and 2,190,445 bp, and the average GC% was 50.56 and 50.66%, respectively, (Table 1, Figure 3). A comparison of genome assemblies showed that *E. rostratum* LWI was larger than *B. zeicola* LWII in terms of genome size and the number of predicted genes.

The repeated sequence of *E. rostratum* LWI was 1,259,007 bp and that of *B. zeicola* LWII was 3,747,661 bp, accounting for 3.70 and 11.63% of the genomeassembly, respectively. These repeats mainly included DNA repeats, long interspersed nuclear elements (LINEs), long terminal repeats (LTRs), and unclassified repeats. The genome of *E. rostratum* LWI had the largest proportion of LTRs at 1.58%, and the proportion of DNA transposons was 0.33%, while the proportion of DNA transposons in *B*. *zeicola* LWII was higher at 3.39%, and the proportion of LTRs was 2.35% (Appendix A, Figure 3). Comparing the repeat sequence results of *E. rostratum* LWI and *B*. *zeicola* LWII, neither *E*. *rostratum* LWI nor *B*. *zeicola* LWII contained short-interspersed elements (SINEs); only *B*. *zeicola* LWII contained LINEs, *B. zeicola* LWII had more unclassified repeats than *E. rostratum* LWI, and only *E. rostratum* LWI contained small RNA. Finally, the G + C content, repeat sequence, LTR, gene density, and gene fragment sizes of *E. rostratum* LWI and *B. zeicola* LWII were visualized. Finally, the G + C content, repeat sequence, LTRs, gene density, and gene fragment size of *B. zeicola* LWII were visualized. An analysis of the differences between the two genomes indicated that the genome size and the number of predicted genes of *E. rostratum* LWI were larger than that of *B. zeicola* LWII, the genome fragment size of some LWI fragments was larger than that of *B. zeicola* LWII, and the LTRs of *E. rostratum* LWI and *B. zeicola* LWII differed (Figure 3).

### 3.3. Gene Prediction and Annotation

Based on homology prediction and de novo prediction methods, we combined different software to identify and integrate protein-coding genes. According to our prediction, *E. rostratum* LWI and *B. zeicola* LWII contained 10,457 and 10,108 protein-coding genes, respectively, with an average length of 551 or 583 bases. We used different databases for annotation, such as the Nr, KEGG, GO, KOG, Pfam, PHI-base, and CAZy databases, to annotate *E. rostratum* LWI and *B. zeicola* LWII (Table 2). 

In the Nr annotation, *E. rostratum* LWI had the highest matching degree with *Setosphaeria turcica* (6943), accounting for 67.92% of the total number of genes predicted by Nr, indicating that *S. turcica* has a close genetic relationship. *B. zeicola* LWII had the highest matching degree with *B. zeicola* (7648), accounting for 75.95% of the total genes predicted by Nr, indicating that *B. zeicola* is closely related to *B. zeicola* LWII. Among the top nine strains with a close genetic relationship to *E. rostratum* LWI, the genetic relationship between *E. rostratum* LWI and *B. zeicola* LWII was not high (363), indicating that the genetic relationship between *E. rostratum* LWI and *B. zeicola* LWII is not close. *E. rostratum* LWI was not included in the top nine strains closest to *B. zeicola* LWII, and the closest *E. rostratum* LWI relative, *S. turcica*, was very low (97), further suggesting that there is no close relationship between *E. rostratum* LWI and *B. zeicola* LWII (Appendix A).

*Exserohilum rostratum* LWI and *Bipolaris zeicola* LWII had 5204 and 4485 sequences, respectively, which were annotated into 25 KOG databases. In addition to some genes with unknown functions, the number of *E. rostratum* LWI annotations in the KOG database was much larger than that of *B. zeicola* LWII. The category with the most *E. rostratum* LWI and *B. zeicola* LWII annotations was “General function prediction only” (1016 and 1079), accounting for 19.52 and 24.06% of the total number of KOG annotations, followed by “Posttranslational modification, protein turnover, chaperones” (482 and 330) and “Translation, ribosomal structure, and biogenesis” (308 and 259) (Figure 4). In the Pfam database, there were 7881 and 7966 protein genes in *E. rostratum* LWI and *B. zeicola* LWII, respectively, which were similar to known proteins in the Pfam database. 

*Exserohilum rostratum* LWI and *Bipolaris zeicola* LWII were annotated using the KEGG database. A total of 10,026 genes were annotated for *E. rostratum* LWI and 10,051 for *B. zeicola* LWII. The top 21 metabolic pathways annotated by *E. rostratum* LWI and *B. zeicola* LWII were very similar. Among the five categories of “Metabolism”, “Genetic Information Processing”, “Environmental Information Processing”, “Cellular Processes”, and “Organismal Systems”, the largest number was in “Global and overview maps” (925 and 820), followed by “Carbohydrate metabolism” (359 and 326). The number of annotations of the rest of the metabolic pathways was not much different, but in some metabolic pathways, the number of annotations of *E. rostratum* LWI was more than that of *B. zeicola* LWII (Figure 5). Except for sequences without subject annotations, the top 10 *E. rostratum* LWI sequences described in KEGG were all part of the glycoside hydrolases (GH) family (GH47, GH3, GH2, GH18, and GH76), followed by “Di-copper center-containing protein”, “tyrosinase”, “aldehyde dehydrogenase”, “amidase”, and “glutathione S-transferase” carbohydrate enzymes are closely related to the pathogenicity of *E. rostratum* LWI. The top 10 of LWII in the KEGG sequence description also included the GH family (GH18, GH47, GH2, GH3, and GH10), followed by “protein-arginine deiminase type-4”, “endo-1”, “aldehyde dehydrogenase”, and “cutinase”. The pathogenicity of *B. zeicola* LWII is not only related to carbohydrate enzymes but also associated with specific proteins.

Using the GO database to annotate the functions of *E. rostratum* LWI (Figure 6) and *B. zeicola* LWII (Figure 7), the 4582 and 5282 annotations of *E. rostratum* LWI and *B. zeicola* LWII were divided into three categories: “cellular components”, “molecular functions”, and “biological processes”. The clusters of *E. rostratum* LWI and *B. zeicola* LWII annotations were similar. The most annotated group of “Cellular component” was “cell and cell part”, the most annotated “Molecular function” was “catalytic activity and binding”, and the most annotated “Biological processes” was “metabolic process and cellular process”.

### 3.4. Prediction and Analysis of Pathogenicity-Related Genes

In the screening of secretory proteins, SingalP [34] was used to identify 1070 proteins containing secretory signals for *E. rostratum* LWI and 893 proteins for *B. zeicola* LWII. Next, the common regions of TMHMM [35] and Phiobius [36] were used to identify 915 proteins without a transmembrane structural domain in *E. rostratum* LWI and 711 in *B. zeicola* LWII. Further, a combination of the WOLF POSR [38] and ProtComp [37] analyses was used to find 504 proteins belonging to the extracellular secretory type for *E. rostratum* LWI and *B. zeicola* LWII, respectively, 420 proteins belonging to the extracellular secretory type for *B. zeicola* LWII, and the remaining 325 and 223 protein sequences for *E. rostratum* LWI and *B. zeicola* LWII with signal peptides but translocated to different organelles or plasma membranes in the cell. Finally, using Pred GPI [39] to remove ankyrins, 494 secreted proteins were found in *E. rostratum* LWI and 382 in *B. zeicola* LWII. The effector protein was then found from the screened secreted proteins. We first screened amino acids with less than 300 amino acids and used SnapGene software to screen more than four cysteine genes. Finally, using EffectorP 3.0, we screened out 164 and 123 effector proteins in *E. rostratum* LWI and *B. zeicola* LWII, respectively. 

*Exserohilum rostratum* LWI and *Bipolaris zeicola* LWII were predicted to have 508 and 447 genes with ≥60% homology in the PHI database, respectively (Figure 8).Among these, *E. rostratum* LWI and *B. zeicola* LWII were the most enriched for “reduced virulence” (295 and 251) and “unaffected pathogenicity” (149 and 137).In other classifications, the difference in the number of *E. rostratum* LWI and *B. zeicola* LWII was not large. For *E. rostratum* LWI and *B. zeicola* LWII, the most critical annotated genes for pathogenicity (hypervirulence) were 12 and 10, respectively.

During the initial stages of infection, pathogens can use CAZymes primarily to degrade the polysaccharide components of the host cell wall [46,47]. We annotated *E. rostratum* LWI and *B. zeicola* LWII using the CAZy database to determine which specific enzymes were associated with the host range and pathogenesis. There were 566 and 517 CAZy annotations for *E. rostratum* LWI and *B. zeicola* LWII, respectively. In general, the difference in the number of genes assigned to the six categories between *E. rostratum* LWI and *B. zeicola* LWII was small. The CAZymes encoding gene models were divided into six major categories, with 54 and 48 carbohydrate esterases (CEs)for *E. rostratum* LWI and *B. zeicola* LWII, respectively, 249 and 241 occupied by glycoside hydrolases (GHs), 251 and 244 by glycosyltransferases (GTs) with 92 and 78, polysaccharide lyases (PLs) with 17 and 18, auxiliary modular enzymes (AAs) with 142 and 123, and carbohydrate-binding modules (CBMs) with 12 and 9, respectively (Figure 9). 

Among the total CAZy annotated in *E. rostratum* LWI and *B. zeicola* LWII, CEs, PLs, and GHs accounted for 57.9% in *E. rostratum* LWI and 59.4% in *B. zeicola* LWII. The GH enzyme family breaks the “glycosidic bond” in carbohydrates or sugars. The number of GHs families in *E. rostratum* LWI and *B. zeicola* LWII was comparable; the most in *E. rostratum* LWI were GH3(16), GH18(13), and GH47(10) and in *B. zeicola* LWII were GH18(13), GH3(11), and GH31(9). The intersection analysis of *E. rostratum* LWI and *B. zeicola* LWII secreted protein genes, PHI genes, and CAZy gene annotation results; there were 8 genes in *E. rostratum* LWI and 13 genes in *B. zeicola* LWII in the intersection part (Appendix A). Combined with the characteristics of the secreted proteins and the annotation results of the two databases, we speculated that the gene IDs of the overlapping parts of *E. rostratum* LWI and *B. zeicola* LWII might be the key pathogenic genes of these two fungi infecting rice.

### 3.5. Phylogenomics Analysis

Clustering gene expression data thus provides important insights into gene coregulation and gene cellular function. A phylogenetic tree was constructed based on the results of single-copy homologous genes identified by gene family clustering, and the phylogenomic relationships between *E. rostratum* LWI and *B. zeicola* LWII and the remaining 20 strains were investigated using *Fusarium graminearum* PH-1 as the outgroup (Appendix A). The genome-wide map of 2072 single-copy orthologue genes shared by the genomes with 18 strains was well supported, and all its branches had bootstrap values of 100, indicating the confidence level of the branch. From the phylogenetic tree, *E. rostratum* LWI and *Exserohilum rostratum* (Genome assembly: GCA_024086065.1) were clustered on one branch, and the support rate was as high as 100%, indicating that *E. rostratum* LWI is extremely closely related to this strain. *B. zeicola* LWII and *Bipolaris zeicola* (Genome assembly: GCA_016906865.1) were clustered on one branch, and the support rate was as high as 100%, indicating that *B. zeicola* LWII is closely related to this strain. The branches of *E. rostratum* LWI and *B. zeicola* LWII were far apart, indicating that their phylogenetic relationship is not close (Figure 10).

## 4. Discussion

In recent years, the impact of rice spikelet rot disease on human health and rice production cannot be underestimated. It is caused by various fungi in China [3]. However, rice spikelet rot disease caused by *Exserohilumrostratum* and *Bipolariszeicola,* investigated in the present study, has rarely been reported in China. *E. rostratum* is a plant pathogen with a wide range of hosts and has a high impact on grasses and Poaceae [48,49] and was first identified in rice in Venezuela [50]. Researchers in Algeria have found that *E. rostratum* is more invasive to maize than *Bipolaris sorokiniana* in pathogenicity tests of *E. rostratum* and *B. sorokiniana* [51]. In humans, the fungus has mainly caused keratitis and skin diseases [8,52]. *E. rostratum* also causes a number of diseases in animals, and a horse in Florida with chronic obstructive rhinitis was identified as the cause of *E. rostratum* [53]. Researchers in Brazil have, for the first time, isolated the pathogen *E. rostratum* in goats with rhinitis, which is unprecedented in goats [54]. At the same time, studies have shown that the *Brn1* gene can help the study of intraspecific variations of *E. rostratum*, and the *Brn1* gene can also identify *E. rostratum* [55]. *B. zeicola* can cause diseases in maize leaves and other tissues, ultimately leading to a reduced maize yield. Researchers from Korea developed species-specific primers for PCR (Bz-F/Bz-R) and recommended this method for rapid and accurate laboratory identification of *B. zeicola* and the diagnosis of maize diseases caused by *B. zeicola* [56,57]. Meanwhile, host-specific toxins produced by *Cochliobolus* species have been shown to enhance the virulence of pathogens [58]. The HC toxin is a non-ribosomal peptide generated by *B. zeicola*, and it induces a high acetylation of histones upon infection with maize types carrying just the susceptible gene [59,60]. Although *B. zeicola* is primarily pathogenic to maize, *B. zeicola* is also pathogenic to other gramineous crops. *B. zeicola* was first found to be pathogenic to barley in Argentina, and the symptoms were similar to those of *B. zeicola* to maize [61]. Egyptian researchers have found that *B. zeicola* can cause wilting, severe rot, and death in rice seedlings [62]. Rice spikelet rot disease caused by multiple fungi had a major impact on the production of rice in recent years and lacks effective preventive measures. However, due to the rapid development of high-throughput sequencing technology, genome sequencing, and the maturity of bioinformatics analysis tools, these techniques are now widely used to study pathogenic fungal pathogenicity and disease resistance. In this study, Illumina and PacBio Sequel were used for the whole-genome sequencing, assembly, and annotation of *E. rostratum* and *B. zeicola*.

The genome size of *E. Rostratum* LWI was 34,053,972 bp, assembled into 16 chromosomes, and the genome *B. zeicola* LWII was 32,215,838 bp, assembled into 15 chromosomes. To further understand the gene’s function, the annotation analysis of *E. rostratum* LWI and *B. zeicola* LWII was performed. Using GO terms, 4582 *E. rostratum* LWI and 5282 *B. zeicola* LWII genes were annotated in total. The protein sequences of *E. rostratum* LWI were mainly annotated in “Biological Process”, with a total of 14,050, and *B. zeicola* LWII were mainly annotated in “Cellular Component”, with a total of 10,574. *E. rostratum* LWI and *B. zeicola* LWII had 10,026 and 10,051 protein genes assigned to the KEGG pathway, respectively. The pathway with the largest proportion was “Global and overview maps”, with 925 in *E. rostratum* LWI and 820 in *B. zeicola* LWII. *E. rostratum* LWI and *B. zeicola* LWII had 5615 and 4485 genes annotated in the KOG database, respectively.

The cell wall of plants is the first barrier to prevent the invasion of pathogenic fungi. In order to successfully invade, pathogenic fungi degrade the cell wall, including upregulating carbohydrate hydrolases and enzymes related to plant cell wall degradation [63,64,65]. Studies have shown that the GH, PL, and CE superfamilies are closely related to pathogenicity. In this study, *E. rostratum* LWI and *B. zeicola* LWII had far more GH families than PL and CE families, and GH3 and GH18 were slightly higher than those of other GH families. GH3 plays a role in promoting the penetration of plant cell walls in the interactions between ascomycetes and plants [66], while GH18 is widely present in fungi, bacteria, insects, plants, and animals, and its role is to promote pathogenic bacteria. It is likely that these enzymes play a critical role in degrading plant cell walls in our two fungi by colonizing, inhibiting host immune responses, and even acting as virulence factors (3). To completely understand how pathogenic fungi degrade cell walls, it is vital to investigate the secretome and secreted effectors that play a role between hypha and host [67,68]. After a comprehensive analysis using various software packages, we identified 494 secreted proteins and 164 effectors proteins in *E. rostratum* LWI and 382 secreted proteins and 123 effector proteins in *B. zeicola* LWII. In the PHI annotation, *E. rostratum* LWI and *B. zeicola* LWII had 508 and 447 proteins associated with pathogenic genes, respectively. Finally, an integrated analysis of the secreted proteins, PHI, and CAZy of *E. rostratum* LWI and *B. zeicola* LWII was carried out. There were 8 and 13 genes in the intersection of the above three results in *E. rostratum* LWI and *B. zeicola* LWII, respectively. These genes are related to the invasion, colonization, and spread of fungi to plants. The disease process is closely related. As a result of these findings, we now have better knowledge of the interactions between *E. rostratum* LWI and *B. zeicola* LWII and rice, which provides more control options. At the same time, we assembled *E. rostratum* LWI and *B. zeicola* LWII at the chromosome level, greatly improving the assembly quality and laying the foundation for subsequent comparative genomics and resequencing.

## 5. Conclusions

In this study, we assembled *E. rostratum* LWI and *B. zeicola* LWII at the chromosomal level and achieved high-quality genomes of organisms, adding to and upgrading their respective genome databases. We investigated the two fungi’s pathogenic causes from the viewpoint of the genome using the whole-genome sequencing analysis; a comparison of functional databases showed that some genes might be crucial for fungus–host interactions. The results promote the pathogenic study of *E. rostratum* and *B. zeicola* and offer important data sources for investigating rice spikelet rot disease.

## Figures and Tables

**Figure 1 jof-09-00177-f001:**
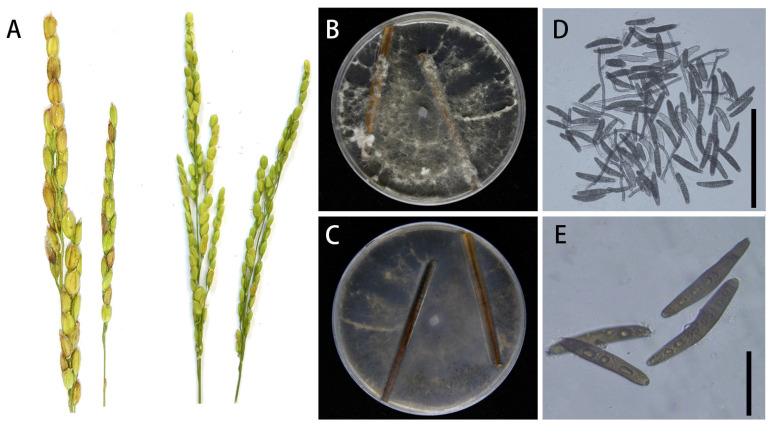
Morphological characteristics of *E. rostratum* LWI. (**A**) Disease symptoms and experimental control group of the *E. rostratum* LWI strain on rice. (**B**,**C**) Growth of *E. rostratum* LWI on TWA-W agar (front and reverse). (**D**,**E**) Conidia produced by *E. rostratum* LWI on TWA-W agar. The (**D**) picture bar = 200 μm, and the (**E**) picture bar = 50 μm.

**Figure 2 jof-09-00177-f002:**
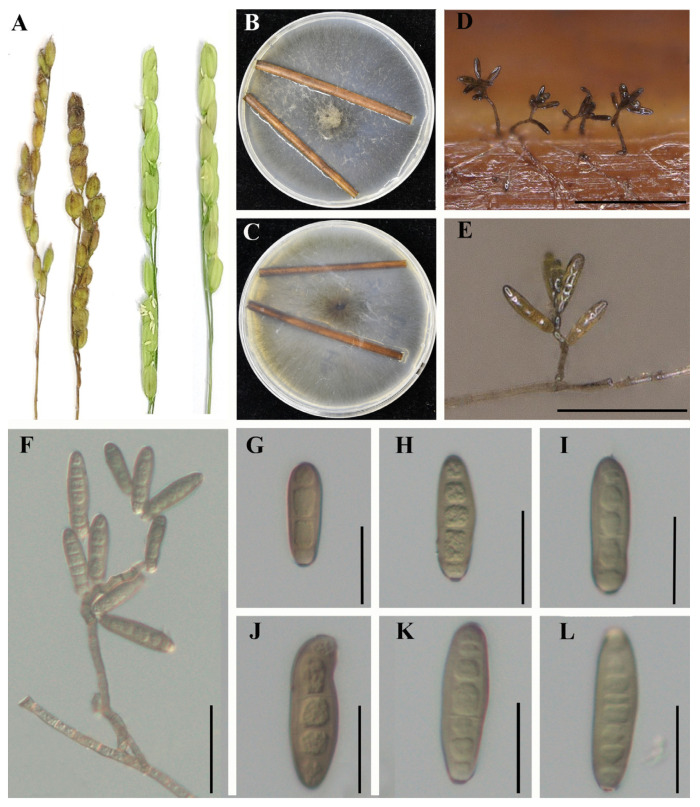
Morphological characteristics of *B. zeicola* LWII. (**A**) Disease symptoms and experimental control group of *B. zeicola* LWII on rice. (**B**,**C**) Growth of *B. zeicola* LWII on TWA-W agar (front and reverse). (**D**,**E**) Conidia produced by *B. zeicola* LWII on wheat straw on TWA-W agar. The (**D**) image bar = 200 μm, and the (**E**) image bar = 100 μm. (**F**–**L**) Conidial structure and morphology of *B. zeicola* LWII. The (**F**) picture bar = 50 μm, and the (**G**–**L**) picture bar = 25 μm.

**Figure 3 jof-09-00177-f003:**
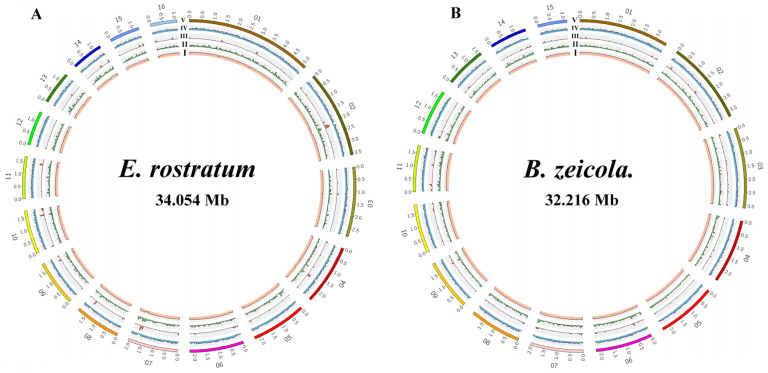
Genome visualization map of partial gene annotation results of *E. rostratum* LWI (**A**) and *B. zeicola* LWII (**B**). **I**: GC content. **II**: Repeat sequence analysis. **III**: Long terminal repeats (LTRs). **IV**: Gene density. **V**: Genomic fragment sizes of *E. rostratum* LWI (**A**) and *B. zeicola* LWII (**B**).

**Figure 4 jof-09-00177-f004:**
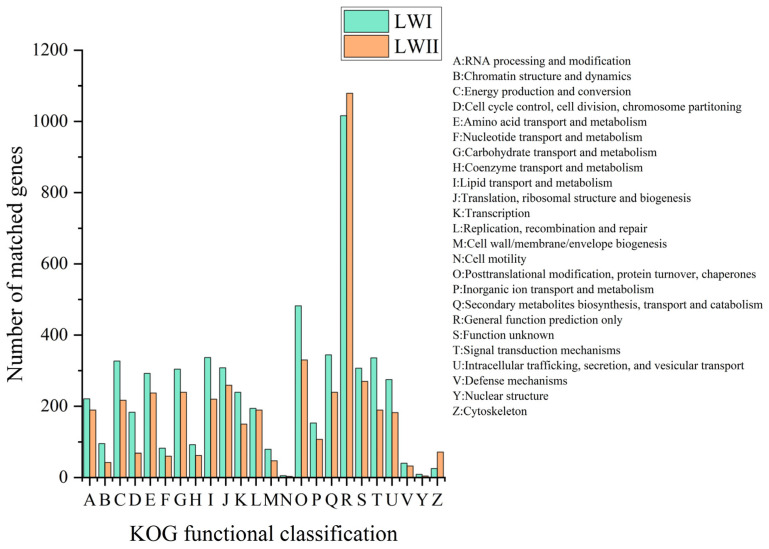
KOG functional classification of the proteins of *E. rostratum* LWI and *B. zeicola* LWII.

**Figure 5 jof-09-00177-f005:**
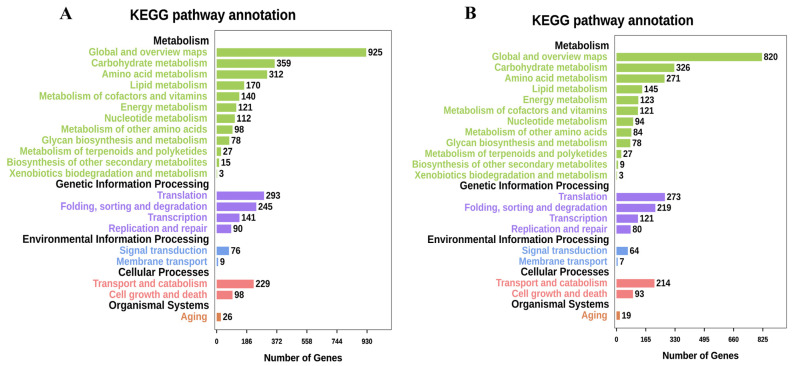
Kyoto Encyclopedia of Genes and Genomes (KEGG) pathway annotation of *E. rostratum* LWI (**A**) and *B. zeicola* LWII (**B**).

**Figure 6 jof-09-00177-f006:**
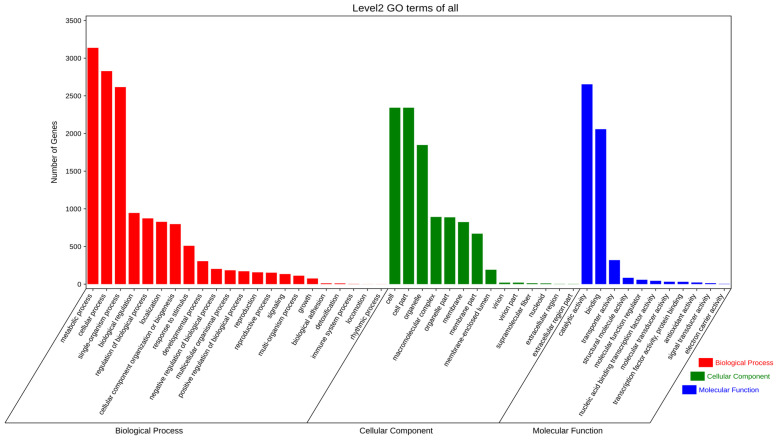
Gene Ontology (GO) functional annotation of *E. rostratum* LWI.

**Figure 7 jof-09-00177-f007:**
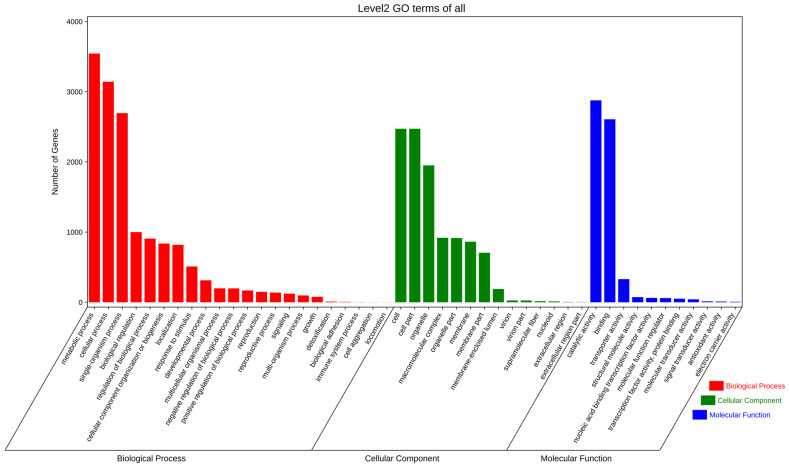
Gene Ontology(GO) functional annotation of *B. zeicola* LWII.

**Figure 8 jof-09-00177-f008:**
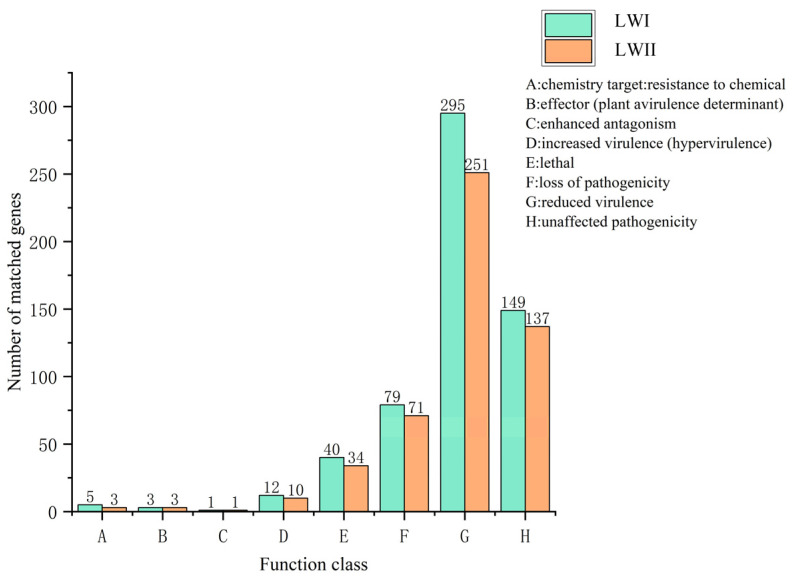
Pathogen–Host Interactions Database (PHI) annotation results of *E. rostratum* LWI and *B. zeicola* LWII.

**Figure 9 jof-09-00177-f009:**
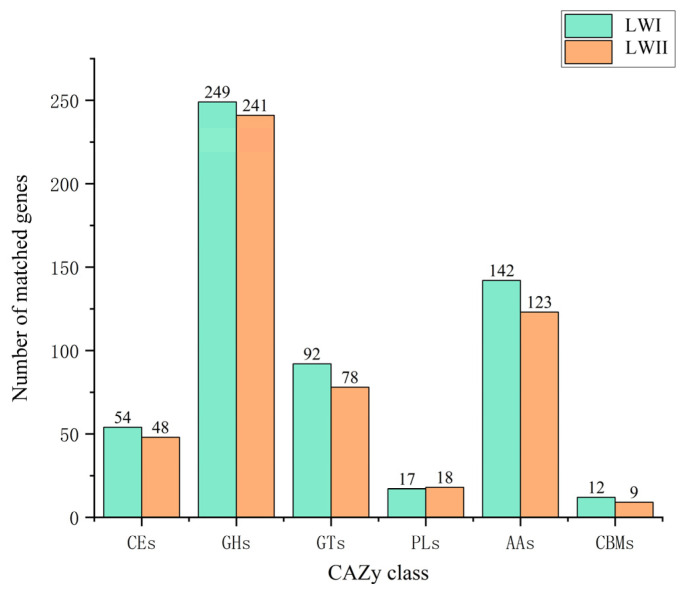
Carbohydrate-active Enzymes Database (CAZy) annotation results of *E. rostratum* LWI and *B. zeicola* LWII.

**Figure 10 jof-09-00177-f010:**
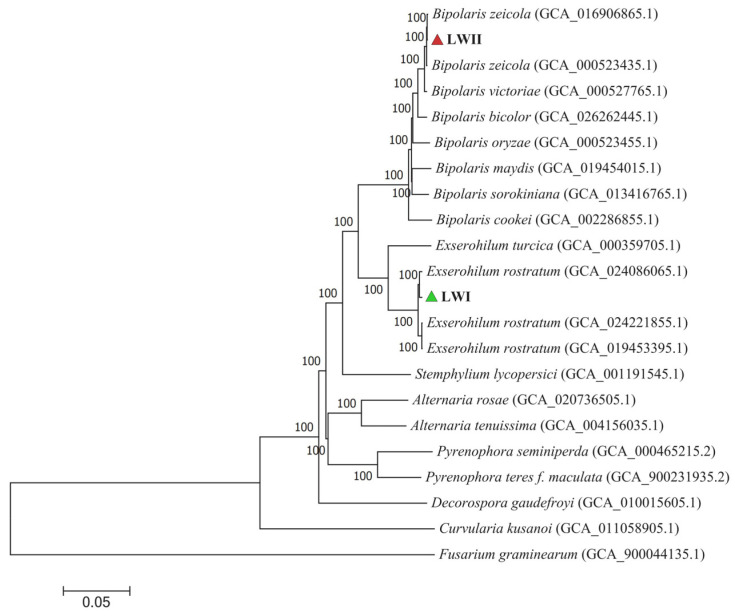
The phylogenetic relationship between *E. rostratum* LWI and *B. zeicola* LWIIwas constructed based on phylogenetic trees of the gene family analysis. There were 2072 single-copy homologous genes. The Bootstrap value indicates the confidence level of the branch in the phylogenetic tree. The triangular symbols indicate the positions of *E. rostratum* LWI (green) and *B. zeicola* LWII (red), and the information in parentheses indicates the gene assembly number.

**Table 1 jof-09-00177-t001:** Genomic features of *E. rostratum* LWI and *B. zeicola* LWII.

Assembly Feature	LWI	LWII
Chromosomes	16	15
Total length (bp)	34,053,972	32,215,838
Longest scaffold length (bp)	5,165,024	3,456,296
Contigs N50 (bp)	2,207,071	2,190,445
Contigs N90 (bp)	1,286,539	1,573,589
Genome coverage	120x	120x
Genome GC %	50.56	50.66
Number of genes	10,457	10,108
Exon average length (bp)	551	583
Exon gene GC%	34.59	33.98
Total gene size (bp)	5,761,019	5,896,621

**Table 2 jof-09-00177-t002:** Statistics of *E. rostratum* LWI and *B. zeicola* LWII gene annotations.

Database	Annotated Gene Number		Annotation Ratio	
	LWI	LWII	LWI	LWII
Nr	10,245	10,079	97.97%	99.71%
GO	4582	5282	43.82%	52.26%
PHI	508	447	0.05%	0.04%
KOG	5204	4485	49.77%	44.37%
KEGG	10,026	10,051	95.88%	99.45%
Pfam	7881	7966	75.42%	78.81%
CAZy	566	517	0.05%	0.05%

Nr, National Center for Biotechnology Information Non-Redundant Protein Database; GO, Gene Ontology; KEGG, Kyoto Encyclopedia of Genes and Genomes; KOG, Eukaryotic Orthologous Groups; Pfam, Database of protein families; PHI, Pathogen–Host Interactions Database; CAZy, Carbohydrate-active Enzymes Database.

## Data Availability

The genome sequence data and assemblies of *E. rostratum* LWI reported in this paper are associated with the NCBI BioProject: PRJNA902501, BioSample: SUB12290528, and Accession Numbers: CP111151–CP111166 in GenBank. The genome sequence data and assemblies of *B. zeicola* LWII reported in this paper are associated with the NCBI BioProject: PRJNA902512, BioSample: SUB12291265, and Accession Numbers: CP111167–CP111181 in GenBank.

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
