# Peer review of "The Chromosome-Scale Genomes of Exserohilum rostratum and Bipolaris zeicola Pathogenic Fungi Causing Rice Spikelet Rot Disease"

_jof, 2023, doi:10.3390/jof9020177_

Round 1
Author Response
Point 1: What is the meaning of LWI and LWII? If species name are there, why need to mention it as LWI and LWII?
Response 1: Thanks for the carefully reading. In the paper, LWI represents the strain number of Exserohilum rostratum, and LWII represents the strain number of Bipolaris zeicola. According to your suggestion, "LWI" has been changed to "Exserohilum rostratum LWI" and "LWII" to "Bipolaris zeicola LWII". (See blue font in the revised text).
Point 2: Kindly mention the accession number of your sequences.
Response 2: Thank you for your reminder. We were really sorry for our careless mistakes. (See blue font, Line 109 to 110). ”Exserohilum rostratum LWI (Accession number: OQ199492) and Bipolaris zeicola LWII (Accession number: OQ199493)”.
Point 3: What was the percentage of repeats/non-repeats in LWII?
Response 3: Thanks for your double-check. In our revised manuscript, "The repeated sequence analysis of LWI and LWII showed that the repeated sequence of LWI was 34,053,972 bp, and that of LWII was 32,215,838 bp" had been corrected to read "The repeated sequence analysis of E. rostratum LWI and B. zeicola LWII showed that the repeated sequence of E. rostratum LWI was 1,259,007 bp, and that of B. zeicola strain LWII was 3,747,661 bp, accounting for 3.70 % and 11.63 % of their genome, respectively". (See blue font, Lines 251, 265 to 267).
Point 4: Line 233: GC length or GC%?
Response 4: Thanks. It is GC%, and the text was revised accordingly. (See blue font, Lines 259).
Point 5: In line no 226, you have mentioned genome size of LWI is 34.909 Mb and LWII is 40.011. Then, how genome size of LWI become larger than LWII in line no 235?
Response 5: Thanks for the carefully reading. Here we first assemble based on Illumina reads, and estimated the size of the two fungi by k-mer analysis. The genome size of E. rostratum strain LWI is 34.909 Mb, and that of B. zeicola strain LWII is 40.011 Mb. Finally, conbined with PacBio Sequel and Illumina data, after high-quality control and assembly, the total genome size of E. rostratum LWI and B. zeicola LWII was finally determined to be 34,053,972 bp and 32,215,838 bp, respectively.
Point 6: When genome coverage of LWII is less than LWI, it is obvious that other elements in the analysis will be less. Please mention, why there is quite a huge gap in the genome coverage?
Response 6: Thanks to the carefully reading and questions. There are many reasons for the large gap in genome coverage. For example, the comparison of different species is very different, the same species, and different samples are very different, and the department and environment of the sample may affect the comparison results. Sample contamination may also cause a significant decrease in the alignment rate and different alignment software and parameter settings may lead to differences in the alignment rate.
Point 7: In Figure 3, please explain the color indications in III & V.
Response 7: Thanks to the carefully reading and the questions raised. The color of Part III in Figure 3 is based on the characteristics of Long terminal repeats (LTRs) to clarify the proportion of LTRs content in the genome and the distribution of chromosomes. The colors used in part V are mainly to distinguish the Genomic fragment sizes of E. rostratum strain LWI and B. zeicola strain LWII. We have added these explanations in the figure legend.
Point 8: Please provide the accession number of the LWI and LWII. It is difficult to cross verify the annotation claim made by the authors without the accession numbers.
Response 8: Thanks for the suggestion, the accession numbers of E. rostratum LWI and B. zeicola LWII have been added in the Data Availability Statement section. (Line 534 to 538).
Point 9: Why genes were clustered to conduct the phylogenetic analysis?
Response 9: Thanks for your question. In genome analysis, the species to be studied and its close relatives will be selected first, and the gene family will be found through comparison. The gene family is derived from the same ancestor, a group of genes composed of multiple copies of a gene through gene duplication. They have obvious similarities in structure and function, and because they are scattered at different positions on the same chromosome or exist on different chromosomes, these genes have different expression regulation patterns. Clustering gene expression data thus provides important insights into gene co-regulation and gene cellular function. In this paper, we use orthologous single-copy genes to construct phylogenetic trees, because orthology originates from the same ancestor gene in evolution and is transmitted vertically. Use the concatenation method (Concatenation) to perform multiple sequence alignments of each single-copy gene between different species, and then connect these aligned single-copy genes end-to-end to form a supergene, and finally use this supergene to construct a phylogeny tree.
Point 10: Based on the clustering, LWI will be Exserohilum sp. And LWII will be Bipolaris species.
Response 10: Thanks to the reviewers for reading carefully. According to the phylogenetic analysis, the LWI strain was identified as Exserohilum rostratum, and the LWII strain was identified as Bipolaris zeicola.
Point 11: When there is differences at the genus level, it is obvious that there will be quite a big differences in the genetic content.
Response 11: Thanks to the reviewers for their careful reading and comments. The GC content of the genome may be affected by the host living environment, sample contamination, etc. Therefore, the GC content of strains LWI and LWII are very close, but the strains LWI and strain LWII belong to different genera after phylogenetic analysis. So we will also use some molecular marker techniques to explore genetic differences in future studies.
Point 12: I am quite surprise to see similar number of xenobiotic degradation and metabolism genes in LWI and LWII despite they are from two different genomes.
Response 12: It is a good question. We suspect that it may have something to do with selective expression of genes by external factors. At the same time, it also provides the direction for us to further study the metabolic correlation between these two strains of fungi.
Point 13: It is suggested that authors must include more number of genome to construct the phylogenetic tree. So that it will give a more better and cleaner picture.
Response 13: Thanks for the suggestion. Bipolaris bicolor ML9021 (GCA_026262445.1), Exserohilum rostratum SR-KPL1 (GCA_024086065.1), Exserohilum rostratum ZM170581 (GCA_024221855.1) and Bipolaris zeicola GZL10 (GCA_016906865.1) have been added to construct the phylogenetic tree. (See Figure 10, Lines 426).
Point 14: The discussion section must include more references.
Response 14: Thanks. We have added more discussion and references to the discussion section. (See blue font, Line 654-655, Lines 658 to 679).
Reviewer 2 Report
With the explosive growth of the population, people's demand for food is also gradually increasing. As the main food crop, rice has always been loved by people. However, the occurrence of diseases has led to the decline of rice yield and quality. Rice spikelet rot caused by E. rostratum and B. zeicola is one of the rice diseases. This study focuses on the whole genome data of the two strains and provides a new way to elucidate their pathogenesis. However, in this manuscript, the author still has some problems. The specific problems are as follows:
1、 As the corresponding phenotype pictures, Figure 1A and Figure 1B should be summarized as one picture, and the author is suggested to make changes and mark them for clear comparison.
2、 The question in Figure 2 is consistent with the above point of view.
3、 The rice spike selected in the phenotypic picture of the article, and the part transferred to the culture medium is straw, so the author is requested to make a supplementary explanation.
4、 In lines 235-236, the author compares the two genomes. It is difficult to understand the relationship between the two genomes and the significance of the comparison. Please explain.
5、 The legend of A and B in Figure 8 are inconsistent, please adjust according to the requirements of the journal.
6、 Most of the references selected in this manuscript are research conducted before 2010. It is recommended that the author consult and summarize the relevant literature review in recent 10 years.
7、 Most of the references selected in this paper are studies conducted before 2010. It is suggested that the author consult and summarize the relevant literature review in the past 10 years, which will make this paper more cutting-edge and innovative.
8、 The format of references is not uniform, so the author should adjust the format according to the requirements of the journal. For example, 507-508 lines.
Author Response
With the explosive growth of the population, people's demand for food is also gradually increasing. As the main food crop, rice has always been loved by people. However, the occurrence of diseases has led to the decline of rice yield and quality. Rice spikelet rot caused by E. rostratum and B. zeicola is one of the rice diseases. This study focuses on the whole genome data of the two strains and provides a new way to elucidate their pathogenesis. However, in this manuscript, the author still has some problems. The specific problems are as follows:
Response : We greatly appreciate your professional comments on our articles. Based on your suggestions, we have revised the previous manuscript, and the specific corrections are given in blue font.
Point 1: As the corresponding phenotype pictures, Figure 1A and Figure 1B should be summarized as one picture, and the author is suggested to make changes and mark them for clear comparison.
Response 1: Thanks to the reviewers for their valuable suggestions. We edited the figure based on your suggestion. (See Figure 1A and Figure 1B, Lines 234, 239).
Point 2: Species names listed for the first time should be full.
Response 2: Thanks. It has been revised. (See blue font, Line 46)
Point 3: The rice spike selected in the phenotypic picture of the article, and the part transferred to the culture medium is straw, so the author is requested to make a supplementary explanation.
Response 3: Thanks to the reviewers for their careful reading, the pathogenic fungi on rice spike were isolated and purified on PDA medium to obtain Exserohilum rostratum strain LWI and Bipolaris zeicola strain LWII. Since sporulation was not found on the PDA medium, in order to allow E. rostratum strain LWI and B. zeicola strain LWII to produce spores, water agar medium with wheat straw (TWA-W) was used. At the same time, relevant literature research was also consulted, which proved that TWA-W medium is the standard medium for sporulation. (Xie, S.N. Molecular systematic studies of Bipolaris and itssimilar genera by multiple gene aggregate analysis, Henan Agricultural University, Zhengzhou, Henan Province, China, 2012.) (See purple font, Line 214).
Point 4: In lines 235-236, the author compares the two genomes. It is difficult to understand the relationship between the two genomes and the significance of the comparison. Please explain.
Response 4: Thanks to the reviewers for their careful reading and questions. We did not perform a comparative genome analysis of E. rostratum and B. zeicola in the manuscript but performed whole genome sequencing and analysis of these two fungi separately. We investigated the two fungi's pathogenic causes from the viewpoint of the genome using whole-genome sequencing analysis; a comparison of functional databases showed that some genes might be crucial for fungal-host interactions. However, during the analysis process, we conducted synteny analysis on the two fungi, but unfortunately, we did not find a linear relationship between the chromosomes of the two fungi.
Point 5: The legend of A and B in Figure 8 are inconsistent, please adjust according to the requirements of the journal.
Response 5: Thanks. We have checked the format of all images and made corrections according to the journal requirements. (See Figure 8 and Figure 9, Lines 379, 394).
Point 6: Most of the references selected in this manuscript are research conducted before 2010. It is recommended that the author consult and summarize the relevant literature review in recent 10 years.
Response 6: Thanks. This manuscript cites a total of 33 references published after 2010 (including 3 articles published in 2010) before supplementing the references. Several references have been added as appropriate based on your suggestions. (See blue and purple font, Line 577 to 578, 654-655, Lines 658 to 679).
“There are few genome assemblies of E. rostratum as a plant pathogen. Currently, only E. rostratum ZM170581 (GCA_024221855.1) isolated from maize has been reported, with a size of 36.34 Mb. This is the first reported genome sequence of E. rostratum isolated from maize”, “Researchers in Algeria have found that E. rostratum is more invasive to maize than Bipolaris sorokiniana in pathogenicity tests of E. rostratum and B. sorokiniana.”, “E. rostratum also causes a number of diseases in animals, and a horse in Florida with chronic obstructive rhinitis was identified as the cause of E. rostratum. Researchers in Brazil have for the first time isolated the pathogen E. rostratum in goats with rhinitis, which is unprecedented in goats. At the same time, studies have shown that the Brn1 gene can help the study of intraspecific variation of E. rostratum, and the Brn1 gene can also identify E. rostratum.”, ”Researchers from Korea developed species-specific primers for PCR (Bz-F/Bz-R) and recommended this method for rapid and accurate laboratory identification of B. zeicola and diagnosis of maize diseases caused by B. zeicola. Meanwhile, host-specific toxins produced by Cochliobolus species have been shown to enhance the virulence of pathogens. HC toxin is a non-ribosomal peptide generated by B. zeicola, and it induces high acetylation of histones upon infection with maize types carrying just the susceptible gene. Although B. zeicola is primarily pathogenic to maize, B. zeicola is also pathogenic to other gramineous crops. B. zeicola was first found to be pathogenic to barley in Argentina, and the symptoms were similar to those of B. zeicola to maize. Egyptian researchers have found that B. zeicola can cause wilting, severe rot, and death in rice seedlings.”
Point 7: Most of the references selected in this paper are studies conducted before 2010. It is suggested that the author consult and summarize the relevant literature review in the past 10 years, which will make this paper more cutting-edge and innovative.
Response 7: Thanks to the reviewers for their careful reading and suggestions. Several references have been added appropriately based on your suggestions and the actual publication of related research on E. rostratum and B. zeicola. (See blue font, (See red font, Line 654-655, Lines 658 to 679).
“Researchers in Algeria have found that E. rostratum is more invasive to maize than Bipolaris sorokiniana in pathogenicity tests of E. rostratum and B. sorokiniana.”, “E. rostratum also causes a number of diseases in animals, and a horse in Florida with chronic obstructive rhinitis was identified as the cause of E. rostratum. Researchers in Brazil have for the first time isolated the pathogen E. rostratum in goats with rhinitis, which is unprecedented in goats. At the same time, studies have shown that the Brn1 gene can help the study of intraspecific variation of E. rostratum, and the Brn1 gene can also identify E. rostratum.”, ”Researchers from Korea developed species-specific primers for PCR (Bz-F/Bz-R) and recommended this method for rapid and accurate laboratory identification of B. zeicola and diagnosis of maize diseases caused by B. zeicola. Meanwhile, host-specific toxins produced by Cochliobolus species have been shown to enhance the virulence of pathogens. HC toxin is a non-ribosomal peptide generated by B. zeicola, and it induces high acetylation of histones upon infection with maize types carrying just the susceptible gene. Although B. zeicola is primarily pathogenic to maize, B. zeicola is also pathogenic to other gramineous crops. B. zeicola was first found to be pathogenic to barley in Argentina, and the symptoms were similar to those of B. zeicola to maize. Egyptian researchers have found that B. zeicola can cause wilting, severe rot, and death in rice seedlings.”
Point 8: The format of references is not uniform, so the author should adjust the format according to the requirements of the journal. For example, 507-508 lines.
Response 8: Thanks. We apologize for our carelessness and have checked the formatting of all references and made corrections as required by the journal. (See Line543 to 691)
Reviewer 3 Report
Dear Authors,
I am glad that I had the opportunity to review the manuscript entitled: "The Chromosome-scale Genomes of Exserohilum rostratum and Bipolaris zeicola Pathogenic Fungi Causing Rice Spikelet Rot Disease" by Ke He, Chenyu Zhao, Manman Zhang, Jinshao Li, Qian Zhang, Xiaoyi Wu, Shan Wei, Xujun Chen, Yong Wang and Cheng Li.
The aim of the research presented by the Authors in this manuscript was to assess the pathogenicity and biological characteristics of the pathogenic fungus (Exserohilum rostratum and Bipolaris zeicola) causing rice spikelet rot disease. Modern techniques of whole-genome sequencing were used in the research. The research results have contributed significantly to the knowledge of this pathogen and will also help in the development of strategies to combat this pathogenic fungus.
In general, the research is conducted properly and the manuscript was well prepared. However, I have some comments that the Authors should take into account at a later stage of preparing the manuscript for publication:
1) Abstract - no concisely formulated hypothesis - please complete.
2) Keywords - please do not use the same words here as in the title of the manuscript. Also, please put the keywords in alphabetical order.
3) Materials and Methods - L 100 - please use the scientific name of the rice and specify the rice cultivar.
4) In the Material and Methods section, please also provide the scientific name of the tested pathogenic fungi.
5) Tables 1 and 2 - please explain the meaning of the abbreviations LWI and LWII. Please remember that all tables and figures should be self-explanatory without having to search the manuscript text for an explanation of abbreviations used therein.
6) Figure 3 - the markings on the genomes are illegible. Please explain the meaning of the abbreviations LWI and LWII.
7) Figure 4 - the description of the X and Y axes and the legend are illegible, please enlarge the font.
8) Figure 5 - please explain the meaning of the abbreviations KEGG, LWI and LWII.
9) Figures 6 and 7 - the description of the X and Y axes and the legend are illegible, please enlarge the font. Please explain the meaning of the abbreviations GO, LWI and LWII.
10) Figure 8 - the description of the X and Y axes and the legend are illegible, please enlarge the font. Please explain the meaning of the abbreviations PHI, CAZy, LWI and LWII.
11) Figure 9 - please explain the meaning of the abbreviations LWI and LWII.
12) Please adapt the manuscript more precisely to the requirements of the template in force in the journal JoF.
In conclusion, I believe that due to the novelty of the research presented in this manuscript, the Editors of the Journal of Fungi should consider publishing this manuscript.
Author Response
Point 1: Abstract - no concisely formulated hypothesis - please complete.
Response 1: Thanks for the deficiencies pointed out by the manuscript, we have added hypothesis in the abstract of the revised version. (See green font, Lines 21, 2 to 28).
Point 2: Keywords - please do not use the same words here as in the title of the manuscript. Also, please put the keywords in alphabetical order.
Response 2: Thanks. We have corrected and sorted the keywords section. (See green font, Lines 32 to 33).
Point 3: Materials and Methods - L 100 - please use the scientific name of the rice and specify the rice cultivar.
Response 3: Thanks to the reviewer for the errors pointed out, it was revised as “Oryza sativa L. Zhonghua 11”. (See green font, Lines 110 to 111).
Point 4: In the Material and Methods section, please also provide the scientific name of the tested pathogenic fungi.
Response 4: Thanks to the reviewers for their suggestions. It was modified as “Exserohilum rostratum LWI (Accession number: OQ199492) and Bipolaris zeicola LWII (Accession number: OQ199493)”. (See blue font, Lines 109 to 110).
Point 5: Tables 1 and 2 - please explain the meaning of the abbreviations LWI and LWII. Please remember that all tables and figures should be self-explanatory without having to search the manuscript text for an explanation of abbreviations used therein.
Response 5: Thanks to the reviewer for pointing out these errors. Sorry for our oversight, we have added the scientific names of these two pathogenic fungi in the titles of Table 1 and Table 2. (See red font, lines 263 and 305).
Point 6: Figure 3 - the markings on the genomes are illegible. Please explain the meaning of the abbreviations LWI and LWII.
Response 6: Thanks to the reviewers for their suggestions. LWI and LWII are the strain names of two pathogenic fungi, E. rostratum and B. zeicola. We have added the scientific names of the two pathogenic fungi to Figure 3 and optimized the genome markers. (See blue font, Lines 283).
Point 7: Figure 4 - the description of the X and Y axes and the legend are illegible, please enlarge the font.
Response 7: Thanks to the reviewers for their suggestions. We have optimized and enlarged the relevant fonts for Figure 4 (Lines 321).
Point 8: Figure 5 - please explain the meaning of the abbreviations KEGG, LWI and LWII.
Response 8: Thanks to the reviewer for the question. The KEGG meaning of our Figure 5 is Kyoto Encyclopedia of Genes and Genomes (see Lines 317). LWI and LWII are the strain names of two pathogenic fungi, E. rostratum and B. zeicola. (See green and blue font, Lines 342 to 343).
Point 9: Figures 6 and 7 - the description of the X and Y axes and the legend are illegible, please enlarge the font. Please explain the meaning of the abbreviations GO, LWI and LWII.
Response 9: Thanks to the reviewers for their suggestions. We have supplemented the GO meanings in Figure 6 and Figure 7 and the scientific names of the two pathogenic fungi. In addition, firstly, the GO annotation result is annotated by the database on the website, and the resulting image is set by the background, and we cannot change it; secondly, if the editor changes the font size of the legend, it will affect the viewing experience of the entire image. I am sorry that your suggestion cannot be adopted, and thank the reviewers for their suggestions on this part. (See red font, Lines 353, 355).
Point 10: Figure 8 - the description of the X and Y axes and the legend are illegible, please enlarge the font. Please explain the meaning of the abbreviations PHI, CAZy, LWI and LWII.
Response 10: Thanks to the reviewers for their suggestions. LWI and LWII are the strain names of two pathogenic fungi, E. rostratum and B. zeicola. We have supplemented the meanings of PHI and CAZy in Figure 8, Figure 9 and the scientific names of the two pathogenic fungi. In addition, the picture is also optimized and the font in the picture is enlarged. (See green and blue font, Lines 380 to 381,Lines 395 to 396).
Point 11: Figure 9 - please explain the meaning of the abbreviations LWI and LWII.
Response 11: Thanks to the reviewers for their suggestions. We have supplemented the scientific names of the two pathogenic fungi in Figure 10. (See blue font, Lines 427).
Point 12: Please adapt the manuscript more precisely to the requirements of the template in force in the journal JoF.
Response 12: Thanks to the reviewers for their careful reading and valuable suggestions, we have checked the manuscript and made corrections to the references and legend formats as required by the JoF journal.
In conclusion, I believe that due to the novelty of the research presented in this manuscript, the Editors of the Journal of Fungi should consider publishing this manuscript.